# β-arrestin mediates communication between plasma membrane and intracellular GPCRs to regulate signaling

Maxwell S. DeNies[1], Alan V. Smrcka[2], Santiago Schnell [1,3,4] & Allen P. Liu [1,5,6,7 ✉]

It has become increasingly apparent that G protein-coupled receptor (GPCR) localization is a master regulator of cell signaling. However, the molecular mechanisms involved in this process are not well understood. To date, observations of intracellular GPCR activation can be organized into two categories: a dependence on OCT3 cationic channel-permeable ligands or the necessity of endocytic trafficking. Using CXC chemokine receptor 4 (CXCR4) as a model, we identified a third mechanism of intracellular GPCR signaling. We show that independent of membrane permeable ligands and endocytosis, upon stimulation, plasma membrane and internal pools of CXCR4 are post-translationally modified and collectively regulate EGR1 transcription. We found that β-arrestin-1 (arrestin 2) is necessary to mediate communication between plasma membrane and internal pools of CXCR4. Notably, these observations may explain that while CXCR4 overexpression is highly correlated with cancer metastasis and mortality, plasma membrane localization is not. Together these data support a model where a small initial pool of plasma membrane-localized GPCRs are capable of activating internal receptor-dependent signaling events.

[1] Cellular and Molecular Biology Program, University of Michigan, Ann Arbor, MI, USA. [2] Department of Pharmacology, University of Michigan Medical School, Ann Arbor, MI, USA. [3] Department of Molecular & Integrative Physiology, University of Michigan Medical School, Ann Arbor, MI, USA. [4] Department of Computational Medicine & Bioinformatics, University of Michigan Medical School, Ann Arbor, MI, USA. [5] Department of Mechanical Engineering, University of Michigan, Ann Arbor, MI, USA. [6] Department of Biomedical Engineering, University of Michigan, Ann Arbor, MI, USA. [7] Department of Biophysics, University of Michigan, Ann Arbor, MI, USA. ✉email: allenliu@umich.edu

W hile extracellular inputs, cell membrane receptors, and resulting transcriptional programs are diverse, many receptor-signaling events converge to a reduced number of signaling hubs. Cellular mechanisms that mediate this process, as well as strategies to control these actions remain outstanding questions. Over the last decade, we have learned that G protein-coupled receptor (GPCR) spatiotemporal signaling is one mechanism used by cells to translate diverse environmental information into actionable intracellular decisions while using seemingly redundant signaling cascades[1]. Extensive research has illustrated that GPCRs elicit distinct signaling events at different plasma membrane micro-domains, as well as endocytic compartments that are important for cell physiology and disease pathogenesis[1–9]. These studies support a model where the location, in addition to magnitude, of a signaling event is important for cellular decision-making. Others have shown that GPCR site-specific posttranslational modifications (PTMs) modulate adaptor protein recruitment, GPCR localization, and consequently receptor-signaling events[10–12]. Together these observations motivated us to reexamine some confounding observations pertaining to the relationship of receptor localization, PTM, and signaling for CXC chemokine receptor 4 (CXCR4).

CXCR4 is a type 1 GPCR that regulates a variety of biological processes, such as cell migration, embryogenesis, and immune cell homeostasis[10,13–16]. It is deregulated in 23 different cancers and overexpression is often correlated with metastasis and mortality[17–21]. However, surprisingly, plasma membrane expression is not correlated with metastasis[21] and in some cancer tumor specimens, as well as cell culture models, samples with poor CXCR4 plasma membrane localization remain responsive to CXCR4 agonist[22–26]. CXCR4 is activated by a highly receptor-specific 8 kDa chemokine, CXCL12 (refs. [27–29]). Unlike β-adrenergic receptors, which have been shown to be activated at intracellular compartments in an OCT3 cationic transporter-dependent mechanism[4,7], endocytic-independent internalization of CXCL12 is unlikely due to its size. Given that receptor activation is dependent on ligand binding or transactivation by another receptor[30,31], the aforementioned observations are confounding, as cells with low plasma membrane CXCR4 remain highly responsive to CXCL12. There are two potential explanations for this observation. Firstly, this could be due to spare receptors on the plasma membrane as it is well established that only a limited number of plasma membrane receptor contribute to signaling[32]. Alternatively, this could be due to activation of intracellular pools of receptors.

## Results

**The UMB2 antibody is sensitive to CXCR4 PTMs**. We began first by investigating the role of CXCR4 localization on receptor-signaling and PTM. To do so, we needed a strategy to robustly detect CXCR4 PTM, as well as a method to modulate receptor localization. We previously established the use of a monoclonal CXCR4 antibody (UMB2) as a robust tool to study CXCR4 PTM[33]. This commercially available antibody is raised against the C-terminus of the receptor, and upon CXCL12 stimulus quickly loses its ability to detect CXCR4 due to receptor PTM[33]. To attempt to identify the specific PTMs responsible, we treated lysates from WT and ubiquitination mutant receptor-expressing cells with phosphatase. While phosphatase treatment ablated AKT S473 phosphorylation, no change in UMB2 detection was observed (Supplementary Fig. 1a, b). Since it has also been reported that CXCR4 is methylated at C-terminal arginine residues[34,35], we tested whether CXCR4 methylation was responsible for the agonist-dependent loss in UMB2 detection. Similar to phosphatase treatment, protein methylation inhibition

did not affect UMB2 detection (Supplementary Fig. 1c). Together these results suggest that the agonist-dependent reduction in UMB2 antibody detection is likely due to a combination of CXCR4 PTMs.

**CXCR4 mutation impacts receptor localization**. To manipulate receptor localization, we generated several mutant receptors that modulate the steady-state distribution of CXCR4 within retinal pigment epithelial (RPE) cells (Fig. 1a). We chose RPE cells to study CXCR4 overexpression because they do not have appreciable endogenous CXCR4 expression, are unresponsive to CXCL12, and were previously established as a cell culture model to study CXCR4 biology[33]. We found that by mutating C-terminal lysine residues to arginine (K3R), CXCR4 plasma membrane localization was reduced by >50% (Fig. 1b, c). While the CXCR4 K3R mutant has been previously used to study CXCR4 degradation[36,37], we found that these mutations caused a drastic change in the spatial distribution of CXCR4. This was due to the unintended creation of an R-X-R motif, which has been shown to increase GPCR retention in the Golgi[38–40]. Indeed, mutating a single residue in the R-X-R motif (i.e., K3R/Q) restored receptor plasma membrane localization to near WT levels (Fig. 1b). Interestingly, while total CXCR4 expression was unchanged by the K3R mutant, the K3R/Q mutant had slightly higher expression compared to WT receptor (Fig. 1c). In accordance with previous literature, K3R mutant receptors partially colocalized with a Golgi compartment marker (Fig. 1d)[38,39]. We also noticed a steady-state population of WT CXCR4 retained at the Golgi (Fig. 1d). Non-plasma membrane-localized CXCR4 has been previously reported[41] and could potentially be due to the presence of a K-X-K motif, which has also been implicated in Golgi protein retention[42–44] or receptor overexpression. It is important to point out that while we observed partial CXCR4 colocalization with the Golgi that is marked by GM130, it is evident from our microscopy results that CXCR4 is present at other intracellular compartments as well (Fig. 1d and Supplementary Fig. 2). To explore this further, we examined WT and K3R CXCR4 colocalization with late and early endosome markers Rab7 and EEA1. Similar to GM120, we found that CXCR4 partially colocalized with both of these markers (Supplementary Fig. 2). This suggests that CXCR4 stably resides at multiple intracellular compartments independent of agonist.

**CXCR4 localization does not impact agonist-induced UMB2 detection**. Having established methods to modulate receptor localization and monitor PTM, we proceeded to investigate the role of CXCR4 plasma membrane localization on CXCL12-dependent AKT S473 phosphorylation. Since CXCL12 is not membrane permeable, we hypothesized that plasma membrane localization is essential for CXCR4 signaling and that reducing receptor plasma membrane expression would decrease CXCL12-dependent AKT phosphorylation. Compared to WT, both mutants had significantly reduced CXCL12-dependent AKT phosphorylation (Fig. 1e). This was expected as mutating biologically relevant residues may affect G protein coupling to CXCR4. However, surprisingly, there was no difference in AKT phosphorylation between high (K3R/Q) and low (K3R) plasma membrane-localized mutant receptors (Fig. 1e, f). In addition, while not explicitly investigated, earlier studies using the K3R mutant also reported that CXCL12-induced ERK1/2 phosphorylation was similar between WT and K3R mutant receptor-expressing cells[37]. Since receptor PTM plays a role in mediating receptor signaling, we investigated the effect of receptor localization on agonist-induced CXCR4 PTM. Although UMB2 detection continued to decrease over a course of 3 h

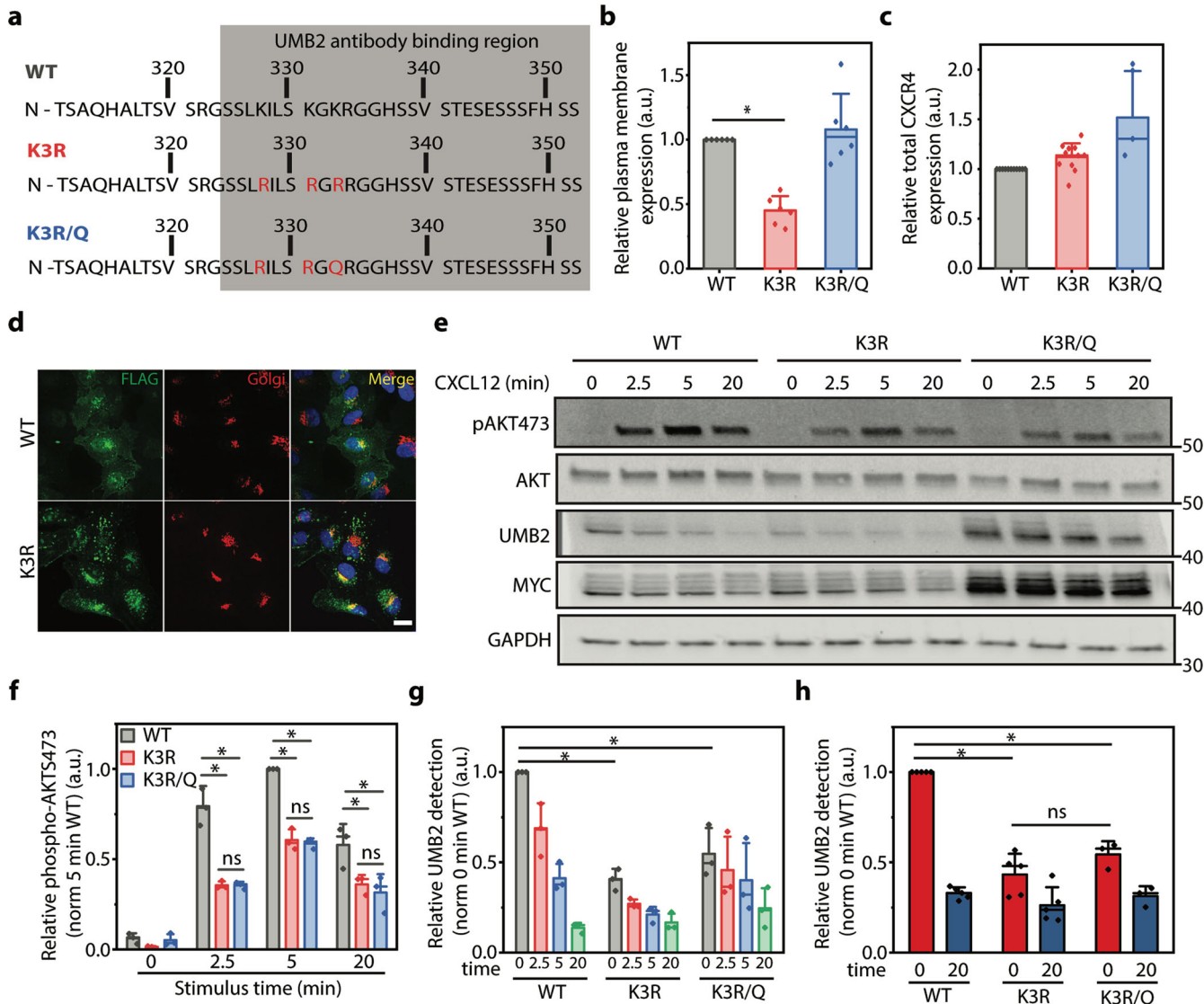

**Fig. 1 CXCL12-depedent AKT S473 phosphorylation and CXCR4 PTM are independent of CXCR4 localization. a** Illustration of CXCR4 mutant receptor constructs. The gray box denotes the binding region of the UMB2 antibody that is sensitive to CXCR4 PTM. **b** Flow cytometry analysis of overexpressed WT and mutant receptor plasma membrane localization in RPE cells. Data were normalized to total receptor and WT plasma membrane expression. **c** Flow cytometry analysis of WT and mutant receptor total expression. Individual data points were normalized to WT CXCR4 expression. **d** Representative microscopy images illustrating the distribution of WT and K3R CXCR4 localization within RPE cells overexpressing each construct. CXCR4 was labeled with a FLAG antibody and the Golgi was detected using a GM130 antibody. Scale bar is 10 μm. Images were captured using 60× magnification on a spinning disk confocal microscope. **e** Representative western blot illustrating CXCL12-induced (12.5 nM) AKT S473 phosphorylation and CXCR4 PTM for WT and mutant receptors. Total CXCR4 was detected using a MYC antibody and unmodified CXCR4 by UMB2. **f** Western blot quantification of AKT S473 phosphorylation for WT and mutant CXCR4. Relative AKT phosphorylation was calculated by normalizing phospho-AKT to total AKT band intensity, and secondly to the 5 min control time point. **g** Western blot quantification of CXCR4 PTM (i.e., UMB2 detection). A decrease in UMB2 detection is correlated to increased CXCR4 PTM. CXCR4 PTM was calculated by dividing the UMB2 intensity by the MYC intensity (total CXCR4) and secondly to the 0 min time point for the WT receptor. **h** Flow cytometry analysis of agonist-dependent WT and mutant CXCR4 PTM. Relative UMB2 detection was determined by dividing median UMB2 detection by total CXCR4 fluorescence and normalized to 0 min WT CXCR4. All experiments were conducted a minimum of three times in RPE cells overexpressing WT or mutant CXCR4. Individual data points from each experiment are plotted; mean, standard deviation (SD), and median line. Statistical significance *$p < 0.05$. Example of flow cytometry gating strategy is shown in Supplementary Fig. 6. Complete raw blots are shown in Supplementary Fig. 7.

(Supplementary Fig. 1d), here we focused on the UMB2 detection in the first 20 min post stimulus, since we were interested in how early CXCR4 PTM regulates cell signaling. We hypothesized that receptor plasma membrane localization is essential for agonist-dependent PTM as agonist-induced receptor PTM is believed to require ligand binding. Surprisingly, irrespective of plasma membrane localization, mutant receptor PTMs were similar (Fig. 1e, g, h). Relative to total receptor expression, initial

detection using the UMB2 antibody for both mutant receptors was also reduced (Fig. 1e, g, h). We believe this is because these mutations occur in the UMB2 antibody-binding region of the receptor (Fig. 1a). Alternatively, this could suggest a difference in steady-state mutant CXCR4 PTM.

Intrigued by the observation of a similar degree of CXCR4 PTM despite vastly different plasma membrane localization, we wondered whether internal (non-plasma membrane) pools of

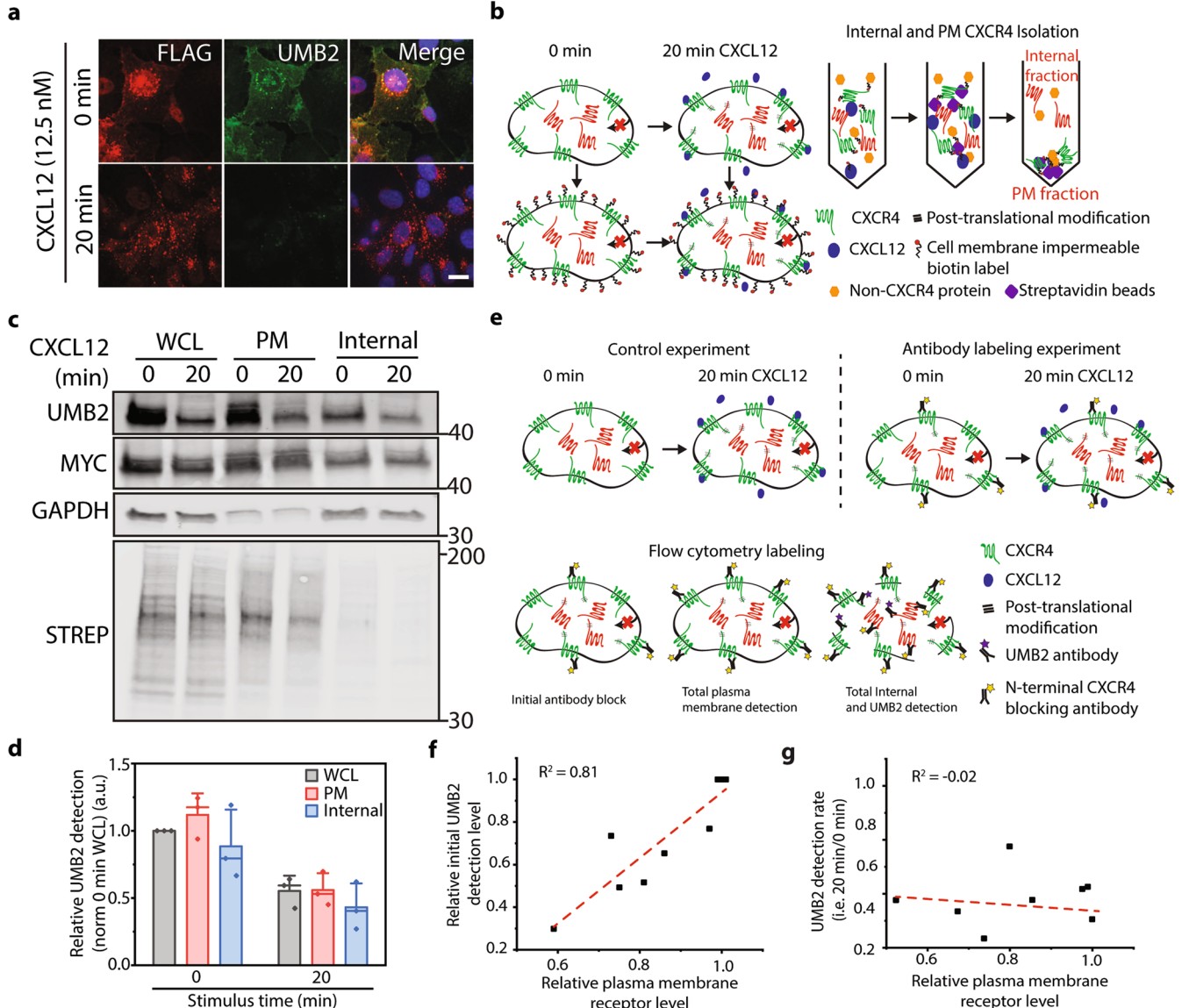

**Fig. 2 Plasma membrane and internal pools of CXCR4 are posttranslationally modified upon receptor stimulus. a** Representative microscopy images of total and non-posttranslationally modified CXCR4 pre and 20 min post CXCL12 (12.5 nM) stimulation. Total CXCR4 is detected by FLAG antibody and non-PTM CXCR4 by UMB2. Images were captured using 60× magnification on a spinning disk confocal microscope. Scale bar is 10 μm. **b** Plasma membrane and internal CXCR4 isolation assay schematic. Receptor internalization was blocked using Dynasore (100 μM) throughout the experiment. At the completion of CXCL12 stimulation, plasma membrane proteins from control and CXCL12-treated samples were covalently labeled, using promiscuous, membrane-impermeable NHS-sulfo-biotin. Afterward, plasma membrane proteins were isolated from whole cell lysate (WCL) by immunoprecipitation and WCL, plasma membrane, and internal pools of CXCR4 were analyzed for PTM by western blot. **c** Representative western blot showing CXCL12-dependent (12.5 nM) CXCR4 PTMs of WCL, plasma membrane, and internal CXCR4. STREP and GAPDH were used as experimental validation. **d** Quantification of CXCR4 PTMs at plasma membrane and internal locations. CXCR4 PTMs were calculated by dividing UMB2 detection (non-PTM CXCR4) by MYC intensity (total CXCR4) and normalized to the 0 min WCL sample. **e** Experimental schematic for antibody blocking experiment. Incubating cells with different concentrations of CXCR4 antibody reduced plasma membrane-localized, ligand-accessible, CXCR4. Afterward, total plasma membrane, total CXCR4, and PTM CXCR4 were quantified by flow cytometry. Experiments were conducted in HeLa cells stimulated with or without 12.5 nM CXCL12. CXCR4 was blocked using various dilutions of a 12G5 allophycocyanin-conjugated CXCR4 antibody. Receptor internalization was blocked throughout all experiments using Dynasore (100 μM). **f** Initial antibody block reduced relative CXCR4 expression and basal UMB2 detection. **g** Relative UMB2 detection 20 min post CXCL12 stimulus plotted against relative CXCR4 plasma membrane expression. $R^2$ values are shown for each experiment. All experiments were conducted in RPE cells overexpressing CXCR4 unless noted. A minimum of three independent replicates were conducted for all experiments and individual data points from each experiment are plotted; mean, SD, and median line. Complete raw blots are shown in Supplementary Fig. 8.

CXCR4 could be posttranslationally modified in response to agonist stimulation and contribute to signaling. To investigate this, we examined the localization of CXCR4 PTM during receptor signaling. As previously observed[33], upon CXCL12 addition UMB2 detection was drastically reduced at both plasma

membrane and intracellular compartments, suggesting that both plasma membrane and internal pools of CXCR4 are posttranslationally modified (Fig. 2a). To test this directly, we developed an assay to selectively isolate plasma membrane proteins from whole cell lysate. We used a membrane-impermeable promiscuous

biotin molecule to selectively label and immunoprecipitate plasma membrane proteins with accessible extracellular domains (Fig. 2b). Receptor internalization was blocked throughout these experiments to keep plasma membrane and internal receptor pools distinct. We hypothesized that only plasma membrane receptors would be posttranslationally modified, as internal pools of receptors are inaccessible to ligand and endocytosis of plasma membrane receptors is blocked. Surprisingly, we found that both surface and internal pools of receptors were posttranslationally modified after ligand addition (Fig. 2c, d). To ensure that the labeling strategy was working as expected, we probed for GAPDH and biotinylated proteins, and showed enrichment in expected localizations (Fig. 2c). To further examine intracellular CXCR4 PTM, we utilized blocking antibodies to effectively tune plasma membrane activity of endogenous CXCR4 in HeLa cells by varying the concentration of blocking antibodies (Fig. 2e). While the blocking antibody effectively reduced steady-state non-posttranslationally modified CXCR4 (Fig. 2f), agonist addition had no effect on CXCR4 PTM after 20 min stimulus irrespective of CXCR4 plasma membrane expression level (Fig. 2g). Together these data support a model where internal pools of CXCR4 are posttranslationally modified in response to CXCL12. In addition, plasma membrane proteins are required for this process as removal of plasma membrane extracellular motifs, using protease treatment completely ablated CXCL12-dependent CXCR4 PTM (Supplementary Fig. 3).

**G$_{\beta\gamma}$ activation is necessary for CXCR4 PTM**. Our findings so far suggest that a signaling cascade may be responsible for intracellular communication between plasma membrane and internal receptor pools. Next, we focused on identifying the proteins responsible for agonist-dependent internal CXCR4 PTM. G proteins are master regulators for GPCR signaling and recent studies have revealed tight spatiotemporal regulation of G protein signaling events[6,7,45]. Upon ligand binding, G protein G$_{\alpha i}$ and G$_{\beta\gamma}$ subunits are released from the GPCR due to guanidine exchange factor activity. G$_{\beta\gamma}$ activates GPCR kinases (GRKs), which quickly phosphorylate the C-terminus of activated receptors leading to β-arrestin recruitment (Fig. 3a)[46]. Therefore, we hypothesized that G$_{\beta\gamma}$ inhibition would reduce CXCL12-induced signaling and consequentially CXCR4 PTM. Indeed, pharmacological inhibition of G$_{\beta\gamma}$ signaling significantly reduced both ERK1/2 and AKT phosphorylation (Fig. 3b–d). In addition, G$_{\beta\gamma}$ inhibition completely ablated CXCL12-dependent CXCR4 PTM (Fig. 3e, f).

**β-arrestin-1 knockdown decreases intracellular CXCR4 PTM**. Since G$_{\beta\gamma}$ activation leads to GPCR phosphorylation and β-arrestin recruitment, we decided to investigate whether β-arrestins were involved in regulating internal CXCR4 PTM. β-arrestins have been previously implicated as potent messenger molecules. Coined signaling at a distance, work from the von Zastrow group proposed a new model for β-arrestin-dependent MAPK signaling, in which β-arrestin-2, activated by stimulated GPCRs on the plasma membrane, traffics to nearby clathrin-coated structures to initiate localized MAPK signaling[8,47]. β-arrestin-1 and 2 are not equal, and significant research has revealed potential site-specific PTM and kinase phosphorylation-specific recruitment to CXCR4, as well as other GPCRs[11]. We hypothesized that β-arrestin-1 or 2 are important for communication between plasma membrane and internal CXCR (Fig. 3a). β-arrestin-1 knockdown led to a reduction in agonist-dependent CXCR4 PTMs while β-arrestin-2 knockdown had no effect (Fig. 4a). β-arrestin-1 knockdown did not affect ERK1/2 phosphorylation, but led to a slight increase in AKT phosphorylation (Fig. 4b–e). This is potentially due to a failure to arrest G protein

signaling. Intrigued by the potential new role of β-arrestin-1 in regulating the communication between plasma membrane and internal pools of receptors, we used the plasma membrane biotinylation assay to determine which CXCR4 population is regulated by β-arrestin-1. While β-arrestin-1 knockdown did not affect plasma membrane-localized CXCR4 PTM, internal CXCR4 PTM was reduced (Fig. 4f–h). Together these data support a mechanism by which G$_{\beta\gamma}$ and β-arrestin-1 work together to regulate communication between plasma membrane and internal pools of CXCR4.

**Intracellular CXCR4 correlates with CXCL12-dependent EGR1 transcription**. Since non-plasma membrane CXCR4 over-expression has been associated with metastatic potential[22,26], we wanted to investigate whether the internal pool of CXCR4 activated distinct signaling pathways compared to plasma membrane-localized receptors. GPCR signaling at intracellular compartments has become increasingly apparent and has been shown to activate different signaling cascades compared to plasma membrane-localized counterparts[1,3,48,49]. Recent work has shown that activation of G protein signaling at the Golgi and endosomes regulates PI4P hydrolysis and *PCK1* transcription, respectively[7,49]. Therefore, we investigated whether activation of intracellular CXCR4 differentially activates downstream signaling compared to plasma membrane receptors. Rather than using mutant receptors (K3R and K3R/Q) that have preexisting signaling defects (Fig. 1), we decoupled the effects of CXCR4 localization from these mutations by removing a synthetic plasma membrane localization sequence commonly used to increase CXCR4 plasma membrane trafficking[36]. By doing so, we were able to monitor agonist-induced CXCR4 PTM and signaling for "WT CXCR4" with high and low plasma membrane expression without mutating biologically relevant C-terminal tail ubiquitination sites (Supplementary Fig 5a). Consistent with our earlier findings using the K3R and K3R/Q mutant receptors, AKT and ERK1/2 phosphorylation, as well as total CXCR4 PTM were not affected by modulating WT CXCR4 localization (Supplementary Fig. 5). Since no overt defect in signaling was observed, we hypothesized that signal location is responsible for differential CXCL12-dependent transcription. To investigate this, we measured early growth response gene 1 (EGR1) transcript levels upon CXCL12 stimulus in RPE cells with high and low plasma membrane CXCR4 expression. EGR1 transcription is downstream of the ERK1/2 pathway and has been shown to be induced by CXCL12 (refs. [50,51]). As expected, WT RPE cells (not over-expressing CXCR4) were unresponsive to CXCL12 (Fig. 5a). However, compared to cells with high plasma membrane expression, cells with low plasma membrane CXCR4 expression had significantly increased CXCL12-induced EGR1 transcript levels (Fig. 5a). This result is inconsistent with the spare receptor model. Furthermore, in agreement with previous work[6,7,49], this suggests that while cells often use some of the same signaling machinery, the localization of a signaling event can lead to different cellular responses. Since β-arrestin-1 plays a role in activating internal CXCR4, we investigated whether inhibition of β-arrestin-1 decreased CXCL12-induced EGR1 transcription. Indeed, β-arrestin-1 knockdown reduced agonist-induced EGR1 transcript levels (Fig. 5b), providing additional evidence that intracellular pools of CXCR4 are physiologically relevant and that their function is dependent on β-arrestin-1.

## Discussion

To the best of our knowledge, this work has revealed a new element of GPCR signaling whereby plasma membrane and internal pools of CXCR4 communicate to regulate cell signaling.

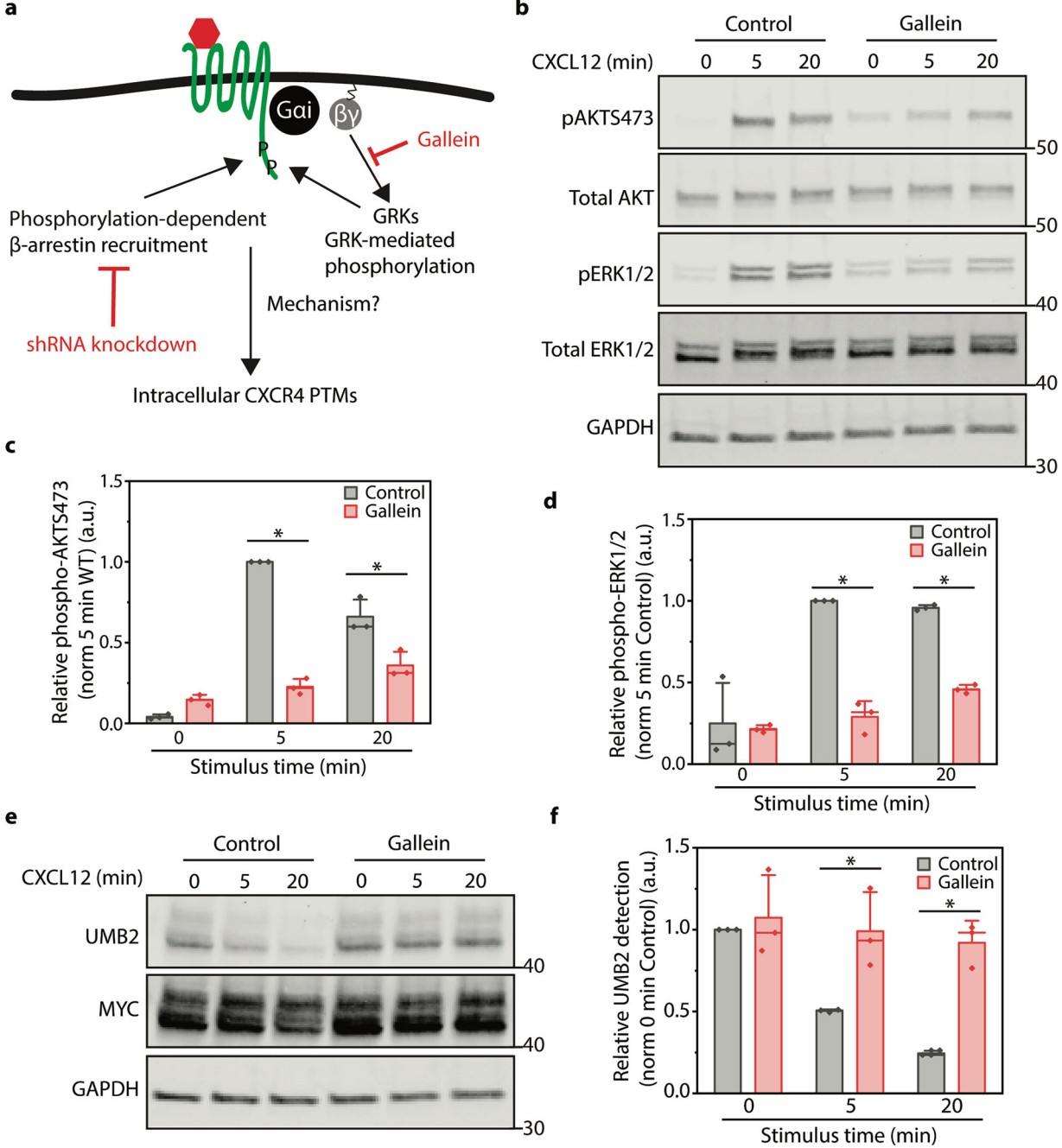

**Fig. 3 G$_{\beta\gamma}$ signaling is essential for CXCR4 signaling and PTM. a** Illustration of the current model of GPCR desensitization. Perturbations used to antagonize different components of the pathway are highlighted in red. **b** Representative western blot illustrating the effects of G$_{\beta\gamma}$ inhibition (Gallein, 10 μM) treatment on CXCL12-induced AKT S473 and ERK1/2 phosphorylation. Cells were pretreated with Gallein for 30 min prior to and throughout each signaling time course. **c, d** Western blot quantification of AKT S473 and ERK1/2 phosphorylation after G$_{\beta\gamma}$ inhibition. Relative signaling protein phosphorylation was calculated by dividing the phosphorylated protein detection by total signaling protein detection and then normalized to the 5 min time point of the control sample. **e** Representative western blot illustrating the effect of G$_{\beta\gamma}$ inhibition (Gallein 10 μM) on CXCR4 PTM. Cells were pretreated with Gallein for 30 min prior to and throughout the signaling time course. **f** Western blot quantification of CXCR4 UMB2 detection (i.e., PTM) upon G$_{\beta\gamma}$ inhibition. CXCR4 PTM was calculated by dividing UMB2 detection (non-PTM CXCR4) by MYC intensity (total CXCR4) and normalized to the 0 min control sample. For all experiments a minimum of three independent replicates were performed. All experiments were conducted in RPE cells overexpressing WT CXCR4 and stimulated with 12.5 nM CXCL12 for the stated time course. Individual data points from each experiment are plotted; mean, SD, and median line. Statistical significance *$p < 0.05$. Complete raw blots are shown in Supplementary Fig. 9.

These observations are distinct from previous work investigating intracellular GPCR signaling, where receptor internalization or OCT3 channel-permeable ligands were required[2–4,7,52,53]. CXCR4 PTM is dependent on G$_{\beta\gamma}$ activation and β-arrestin-1 plays a specific role in regulating intracellular CXCR4 PTM. This

work expands upon the β-arrestin signaling at a distance concept and supports a model, where β-arrestins are not only able signal at a distance at the plasma membrane, but also regulate communication between plasma membrane and internal GPCR populations to influence agonist-dependent transcriptional

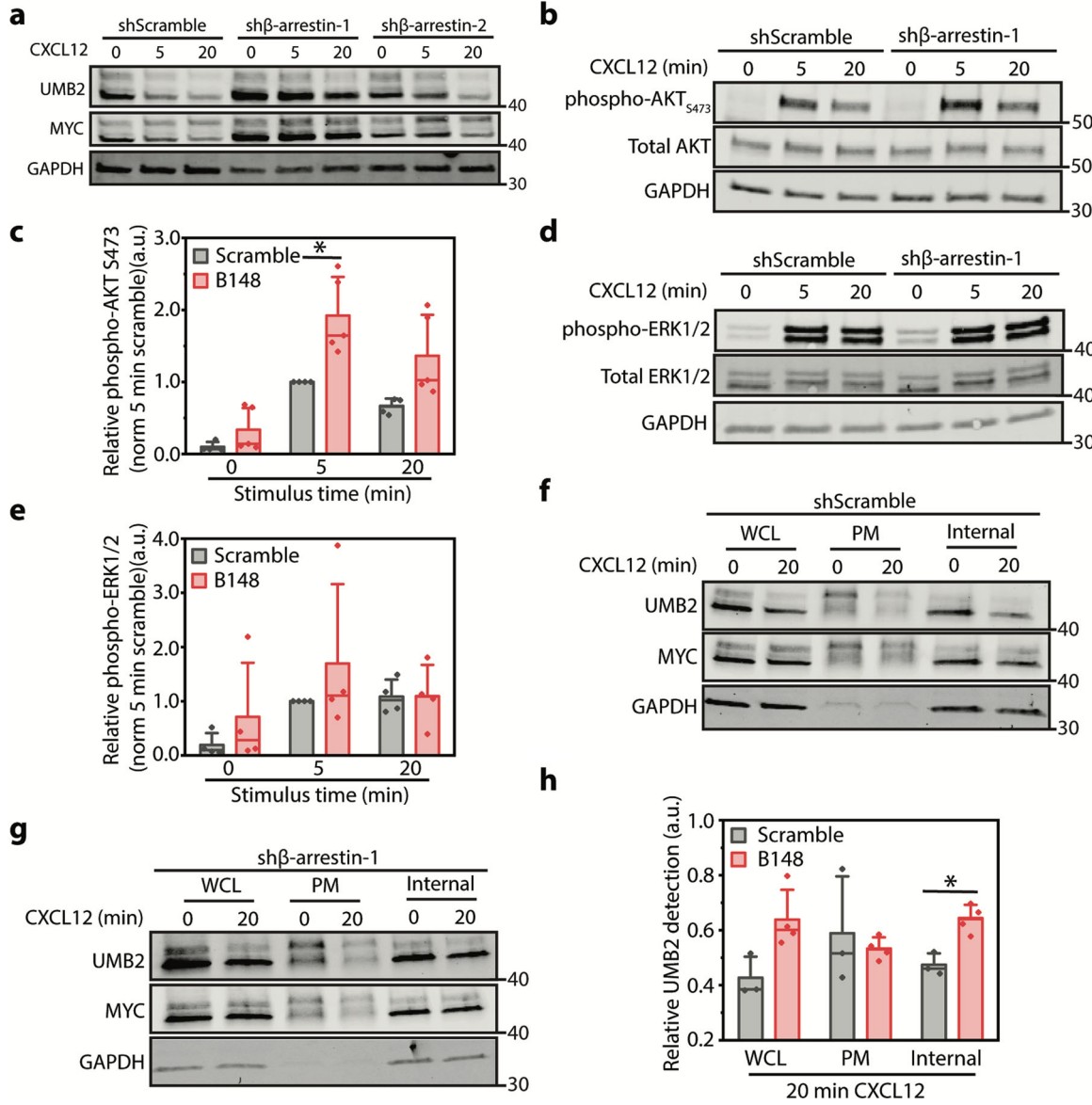

**Fig. 4 β-arrestin-1 regulates agonist-induced internal CXCR4 PTM. a** Representative western blot illustrating the effect of β-arrestin-1 and 2 knockdown on CXCR4 PTM. Relative shRNA knockdown efficiency is shown in Supplementary Fig. 4a–c. **b** Representative western blot illustrating the effect of β-arrestin-1 knockdown on CXCL12-dependent AKT S473 phosphorylation. **c** Western blot quantification of CXCL12-dependent AKT S473 phosphorylation upon β-arrestin-1 knockdown. Data were normalized to phospho-AKT:total AKT and to 5 min normalized control shRNA sample. **d** Representative western blot illustrating the effect of β-arrestin-1 knockdown on CXCL12-dependent ERK1/2 phosphorylation. **e** Western blot quantification of CXCL12-dependent ERK1/2 phosphorylation upon β-arrestin-1 knockdown. Data were normalized to phospho-ERK1/2:total ERK1/2 and to 5 min normalized control shRNA sample. **f**, **g** Representative western blots illustrating total, plasma membrane, and internal pools of CXCR4 PTM upon either scramble or β-arrestin-1 shRNA knockdown. **h** Quantification of CXCR4 PTM at plasma membrane and internal locations upon β-arrestin-1 knockdown. CXCR4 PTMs were calculated by dividing UMB2 detection (non-posttranslationally modified CXCR4) by MYC intensity (total CXCR4) and normalized to the 0 min time point at each location. For all experiments, a minimum of three independent replicates were performed. All experiments were conducted in RPE cells overexpressing WT CXCR4 and stimulated with 12.5 nM CXCL12 for the stated time course. β-arrestin-1 knockdown experiments were conducted using two validated shRNAs (Supplementary Fig. 4). Individual data points from each experiment are plotted; mean, SD, and median line. Statistical significance *$p < 0.05$. Complete raw blots are shown in Supplementary Fig. 10.

programs. A model for communication between plasma membrane and intracellular pools of CXCR4 and its ramification on signaling is summarized in Fig. 5c. There are several potential mechanisms for how this communication may occur that warrant additional research.

Many new questions regarding the molecular mechanism and physiological relevance of plasma membrane and intracellular CXCR4 activation remain unanswered. A limitation of our surface biotinylation approach to study intracellular CXCR4 PTMs is

that it is unable parse out which subpopulation(s) of CXCR4 are being posttranslationally modified. While agonist-induced internal CXCR4 PTM appear to partially occur at the Golgi (Fig. 2a), CXCR4 may also be posttranslationally modified at other intracellular compartments. Understanding which intracellular CXCR4 populations are posttranslationally modified is important for understanding which receptor pool is responsible for CXCL12-induced EGR1 transcription. Furthermore, neither β-arrestin or G$_{\beta\gamma}$ are believed to actively posttranslationally modify

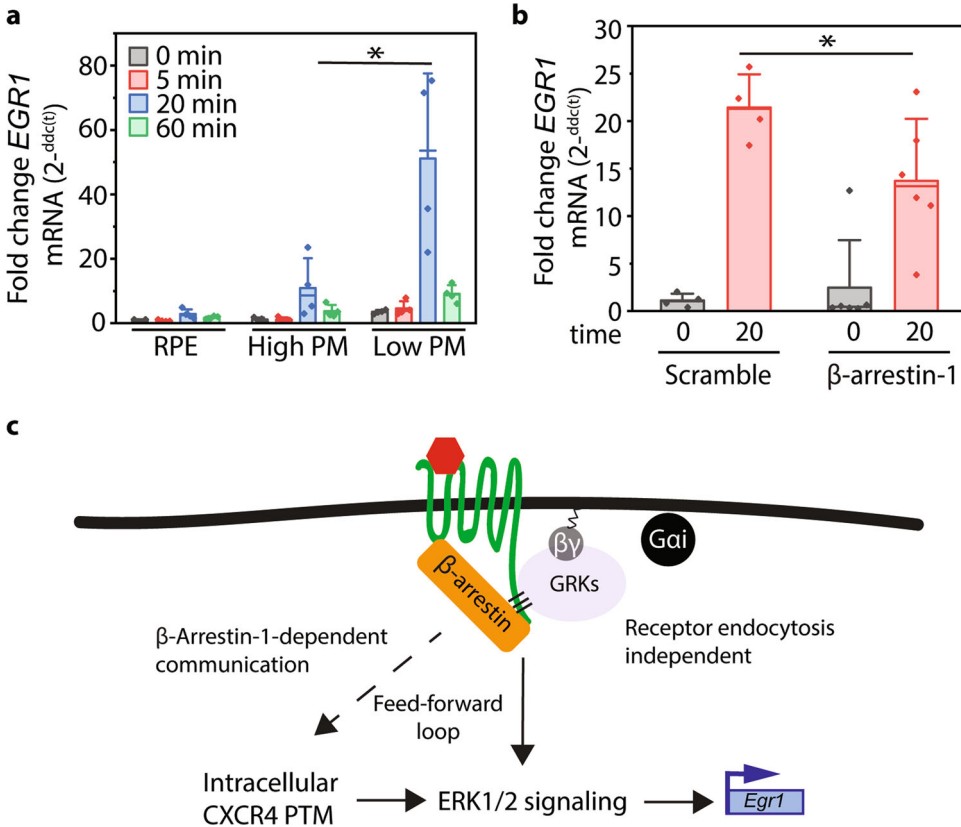

**Fig. 5 Intracellular pools of CXCR4 are primarily responsible for EGR1 transcription. a** qPCR analysis of EGR1 transcription in WT RPE or RPE cells overexpressing high or low plasma membrane-localized WT CXCR4. EGR1 transcript levels were calculated using the ΔΔCT method normalized to GAPDH and 0 min high plasma membrane CXCR4. **b** β-arrestin-1 knockdown reduces CXCL12-dependent EGR1 transcript levels in RPE cells overexpressing high plasma membrane WT CXCR4. EGR1 transcript levels were calculated using the ΔΔCT method. **c** Schematic summarizing a potential model for communication between plasma membrane and internal GPCR pools. All experiments were conducted in RPE cells overexpressing WT CXCR4 and stimulated with 12.5 nM CXCL12 for the stated time course unless noted. β-arrestin-1 knockdown experiments were conducted using two validated shRNAs (Supplementary Fig. 4). Individual data points from each experiment are plotted; mean, SD, and median line. Statistical significance *$p < 0.05$.

proteins. However, $G_{\beta\gamma}$ is a potent kinase activator therefore identification of the kinase or other protein machinery responsible for internal CXCR4 PTMs is necessary. Interestingly, $G_{\beta\gamma}$ has been shown to traffic to intracellular compartments, including the Golgi after agonist activation, independent of receptor endocytosis[54,55]. Understanding the specific PTMs of plasma membrane and internal CXCR4 populations could also provide important insights pertaining to the function and fate of activated intracellular receptors. It is possible that this mechanism may regulate receptor trafficking to the plasma membrane, effectively providing cells with a short-term memory of prior signaling events.

It is now widely accepted that GPCRs signal from multiple intracellular localizations and that the localization of GPCR activation differentially impacts downstream signaling. Consequently, it is plausible that aberrant signaling from different cellular compartments is associated with disease. Thus, inhibiting receptor signaling at different cellular compartments may be an effective therapeutic strategy that reduces side effects. While there has been great interest over the last few decades to modulate CXCR4 function in both HIV and cancer therapeutics research, therapeutic success has been limited[56]. AMD3100 (Plerixafor) is the only CXCR4 inhibitor that, to our knowledge, is FDA approved[57]. This inhibitor is membrane permeable and ubiquitously inhibits CXCL12-induced signaling by preventing interactions between CXCL12 and CXCR4 (ref. [57]). Since CXCR4 plays an important role in multiple cellular processes, the ability

to selectively target different populations of CXCR4 might lead to more successful therapies that do not act as a blunt instrument, which ubiquitously disrupt downstream signaling activity from the inhibited receptor. Similarly, significant recent research has led to the development of biased peptide agonists and antagonists for CXCR4, as well as other GPCRs[56]. The goal of these therapies is to selectively modulate receptor-signaling pleiotropism[56]. To our knowledge, these efforts have largely focused on modulating signaling of plasma membrane receptors. However, coupled with our findings, it would be interesting to explore whether intracellular targeting can improve their therapeutic utility. Additional research exploring localized receptor signaling is necessary to truly elucidate the relationship between localization and signaling. We believe this new paradigm has the potential to influence the design of next-generation therapies.

While many questions remain, the data presented expand the role of $G_{\beta\gamma}$ and β-arrestins in regulating GPCRs, and support a new model of GPCR signaling whereby plasma membrane and internal pools of receptors communicate to collectively determine a cellular response. These observations may resolve the paradox that while CXCR4 overexpression is associated with metastatic potential, plasma membrane localization is not. In addition, this work supports a growing amount of evidence supporting that targeting specifically intracellular GPCR populations or the downstream signaling cascades activated by these pools might lead to improved therapeutic strategies for treating cancer, cardiovascular disease, and pain management[6,7,40].

**Table 1 Reagents.**

| | PN | Supplier | Working concentration |
|---|---|---|---|
| **Inhibitors** | | | |
| Gallein | 3090/50 | R&D Systems | 10 µM |
| Dynasore | 324410 | Sigma | 100 µM |
| MTA | D5011 | Thermo Fisher | 200 µM |
| **Antibodies** | | | |
| ms-FLAG-647 | A01811-100 | Genscript | 1:1000 (FC/IF) |
| rb-MYC | A190-105A | Bethyl Laboratories | 1:5000 (WB) |
| rb-CXCR4 (UMB2) | Ab124824 | Abcam | 1:2000 (WB), 1:1000 (FC/IF) |
| Rb-phospho-ERK1/2 | 4370 S | Cell Signaling Technologies | 1:2000 (WB) |
| ms-total-ERK1/2 | 4696 S | Cell Signaling Technologies | 1:1000 (WB) |
| rb-phospho-AKT S473 | 4060 S | Cell Signaling Technologies | 1:2000 (WB) |
| rb-total AKT | C67E7 | Cell Signaling Technologies | 1:1000 (WB) |
| rb-GM130 | 12480 S | Cell Signaling Technologies | 1:1000 (IF) |
| ms-GAPDH | sc-47724 | Santa Cruz Biotechnology | 1:1000 (WB) |
| STREP-568 | S11226 | Thermo Fisher | 1:5000 (WB) |
| ms-CXCR4-APC (12G5) | FAB170A | R&D Systems | 1:200–1:2000 (FC) |
| rb-β-arrestin-1 | 30036 S | Cell Signaling Technologies | 1:1000 (WB) |
| gt-anti-rb Dylight 800 | SA5-35571 | Thermo Fisher | 1:5000 (WB) |
| gt-anti-ms Dylight 680 | 35518 | Thermo Fisher | 1:5000 (WB) |
| gt-anti-rb-488-AlexaFluor-Plus | A32731 | Thermo Fisher | 1:1000 (FC/IF) |
| Gt-anti-rb-555 | 84541 | Thermo Fisher | 1:1000 (IF) |
| Gt-anti-rb-Alexafluor-Plus | A32733 | Thermo Fisher | 1:1000 (IF) |
| Gt-anti-ms-488 | A10680 | Thermo Fisher | 1:1000 (IF) |
| Gt-anti-ms-647 | A21235 | Thermo Fisher | 1:1000 (IF) |
| Rb-anti-EEA1 | 3288 S | Cell Signaling Technologies | 1:200 (IF) |
| Rb-anti-Rab7 | 9367 S | Cell Signaling Technologies | 1:100 (IF) |
| **Biologics** | | | |
| CXCL12 | 350-NS-050 | R&D Systems | 12.5 nM |
| Pronase | 10165921001 | Sigma | 0.1% Solution |
| Lambda phosphatase | sc-200312A | Santa Cruz Biotechnology | Manual instructions |
| **Others** | | | |
| Streptavidin agarose | 20361 | Thermo Fisher | |
| DAPI | D9542 | Sigma | 1 µg/ml |
| NewBlot PVDF Stripping buffer | 928–40032 | LiCor | Manual instructions |
| PVDF 0.22 µm membranes | IB401001 | Thermo Fisher | NA |
| NHS-Sulfo-LC-biotin | 21335 | Thermo Fisher | 1 mg/ml |
| ITAQ Universal SYBR Green | 1725121 | BioRad | Manual instructions |
| iScript cDNA Synthesis Kit | 1706891 | RioRad | Manual instructions |
| Quick-RNA miniprep | R1054 | Zymo Research | Manual instructions |

## Methods

**Equipment**. LiCor Odessey CLX & SA Imagers, Azure Sapphire 4 laser Imager, and BioRad RT-qPCR ThermoCycler.

**Cell culture**. HeLa cells were originally obtained from ATCC. HeLa cells were cultured in DMEM media (Corning) supplemented with 10% FBS (Corning). RPE cells were a gift from Dr. Sandra Schmid at UT Southwestern. All stable cell lines were directly derived from this RPE line. RPE cell lines were cultured in DMEM/F12 media (Corning) supplemented with HEPES, glutamate, and 10% FBS (Corning). HEK293T cells were obtained from ATCC and grown in DMEM (Corning) media supplemented with 10% FBS.

**DNA constructs and stable cell lines**. WT CXCR4 was generated as previously described[33]. K3R and K3R/Q mutant receptors were generated by PCR mutagenesis of WT CXCR4 in the pLVX plasmid, using the NEB Quick-change mutagenesis kit. The low plasma membrane CXCR4 construct was generated by PCR amplification (excluding the 5′ plasma membrane HA localization peptide) and restriction enzyme cloning, using the BsrRI and EcoRI restriction enzymes. All CXCR4 constructs had an N-terminal FLAG tag and C-terminal MYC tag for easy antibody detection. Stable cell lines expressing WT and mutant CXCR4 receptors were generated by lentiviral transduction. Lentiviruses (shRNA and CXCR4 constructs for stable cell lines) were generated by co-transfecting HEK293T cells with the pLVX transfer plasmid, psPAX2, and pMD2.G lentiviral envelope and packaging plasmids. To generate stable cell lines, supernatant media containing mature lentiviral particles was collected 4 days post transfection and added to RPE cells, and cells stably expressing the constructs were generated via puromycin selection (3 µg/ml). All transfections were conducted using Lipofectamine 2000 (Life technologies).

**Flow cytometry experiments**. Flow cytometry experiments for plasma membrane receptor labeling were conducted, as previously described[33]. For intracellular staining, cells were first disassociated using 50 µM EDTA in $Ca^{2+}$-free PBS and fix for 10 min in 4% paraformaldehyde at room temperature. Afterward, cells were permeabilized using 0.2% Triton-X 100 for 10 min at room temperature. Intracellular targets were labeled with primary antibodies for 1 h at room temperature after which cells were washed with PBS and incubated with secondary antibodies for 1 h at room temperature—see Table 1 for antibody specifics. Afterward, cells were washed 1× with PBS and 25,000 events were analyzed by the Guava EasyCyte flow cytometer for each experimental condition. When co-staining, compensation was conducted post experiment using controls with either 488 or 640 fluorescence alone. After fluorescence compensation, the median fluorescence was calculated for each channel and sample, as well as for no stain and RPE WT controls (not expressing CXCR4). As previously described, median control sample fluorescence was subtracted from each sample, and data were normalized and plotted, as described in each figure legend[33]. Representative flow cytometry data collected can be found in Supplementary Fig. 6.

**Cell signaling, shRNA and inhibitor experiments**. Cells were seeded in 12-well plates 24 h prior to each signaling experiment achieving 70–80% confluence at the experimentation time. Cells were serum-starved in DMEM/F12 media without FBS for 4 h prior to each signaling experiment. For inhibitor experiments (Gallein, Dynasore), cells were pretreated for 30 min with the respective inhibitors and throughout the signaling experiment. For shRNA experiments, cells were transduced with either scramble or β-arrestin-1 or 2 shRNA (Table 2) for 3 days. shRNA lentiviral particles were generated, as described above. Afterward, cells were stimulated with 12.5 nM CXCL12 (R&D Systems) for the labeled time course. Samples were washed with PBS 1× and lysed, using RIPA buffer supplemented with protease inhibitors (EDTA-free Peirce protease inhibitor cocktail) and phosphatase

**Table 2 shRNA sequences.**

| shRNA (pLKO1 vector) | Sequence |
|---|---|
| Scramble (nontargeting) | Sigma-SCH002 |
| β-arrestin-1 | |
| TRCN0000230148 | CCGGAGATCTCAGTGCGCCAGTATGCTCGAGCATACTGGCGCACTGAGATCTTTTTTG |
| TRCN0000219075 | GTACCGGACACAAATGATGACGACATTGCTCGAGCAATGTCGTCATCATTTGTGTTTTTTTG |
| β-arrestin-2 | |
| TRCN0000280686 | CCGGGATACCAACTATGCCACAGATCTCGAGATCTGTGGCATAGTTGGTATCTTTTTG |
| TRCN0000280619 | CCGGGCTAAATCACTAGAAGAGAAACTCGAGTTTCTCTTCTAGTGATTTAGCTTTTTG |

**Table 3 RT-qPCR primer sequences.**

| Primers | Forward | Reverse |
|---|---|---|
| GAPDH | GAGTCAACGGATTTGGTCGT | CTTGATTTTGGAGGGATCTCGC |
| EGR1 | GGTCAGTGGCCTAGTGAGC | GTGCCGCTGAGTAAATGGGA |
| β-arrestin-1 | ATCCCTCCAAACCTTCCATG | TGACCAGACGCACAGATTTC |
| β-arrestin-2 | AAGTGTCCTGTGGCTCAA | TTGGTGTCCTCGTGCTTG |

inhibitors (HALT phosphatase inhibitor). For Lambda phosphatase experiments, phosphatase inhibitors were excluded in the lysis buffer. After incubating cells with lysis buffer for 10 min on ice, lysates were collected and centrifuged at $16,000 \times g$ for 45 min at 4 °C. Afterward, lysates were immediately stored at −20 °C or processed for immunoprecipitation or western blotting.

**Immunoprecipitation**. For plasma membrane biotinylation experiments, biotinylated plasma membrane proteins were isolated from WCL using high capacity streptavidin agarose beads (Table 1). Approximately 35 μl of bead slurry was added to 350 μl WCL and incubated overnight (~18 h) rotating at 4 °C. Afterward, samples were pelleted (centrifuged for 3 min at $2000 \times g$) and internal (non-plasma membrane) proteins collected by removing the supernatant. To prevent potential biotinylated protein contamination, only 200 μl of supernatant was removed. Afterward, beads were washed three times with RIPA buffer containing protease inhibitors. After the final wash, all buffer was removed.

**Western blotting and data analysis**. Prior to western blotting, samples were incubated with Laminelli buffer supplemented with β-mercaptoethanol (loading buffer). For surface biotinylation samples, β-mercaptoethanol concentration was increased twofold and samples were incubated at room temperature in the loading buffer for 30 min prior to western blotting to denature proteins from beads. Samples were run on SDS–PAGE 4–20% BioRad gels (15 well/15 μl or 10 well/50 μl gels). For all signaling experiments, 12.5 μl of lysate was loaded while for surface biotinylation assays, 35 μl of lysate was loaded. SDS–PAGE gels were run at constant 140 V for ~60 min. Afterward, proteins were transferred to PVDF membranes using the iBlot transfer systems (mixed range proteins 7 min setting) and membrane incubated in blocking solution (1% BSA in TBST) rocking for 1 h at room temperature. Afterward, blots were incubated with their respective antibodies (Table 1) overnight at 4 °C. Prior to secondary labeling, blots were washed three times for 5 min per wash with TBST. Blots were then incubated with the corresponding secondary antibody (Table 1) for 1 h at room temperature. Blots were then washed with TBST as described above. Western blots were dried and imaged using a LiCor Odessey SA, LiCor CLX, or Azure Biosystems Sapphire System. Data were analyzed using the LiCor image studio software to calculate band intensity, as previously described[33]. Specific normalization procedures for each experiment are described in the respective figure legends. All statistics were calculated using two-tailed $t$ tests. Complete western blots shown in Supplementary Figs. 1, 3, and 4 are shown in Supplementary Fig. 11.

**RT-qPCR experiments**. Cells were seeded in six-well plates 24 h prior to each signaling experiment achieving 70–80% confluence at the experimentation time. Cells were serum-starved in DMEM/F12 media for 4 h prior to each signaling experiment. Afterward, cells were stimulated with 12.5 nM CXCL12 (R&D Systems) for the respective time courses shown in the figure legends. RNA was extracted using the Zymogen RNA extraction kit (R1054) and 1 μM cDNA was synthesized using the iScript synthesis kit (BioRad). qPCR assays were conducted using SYBR Green (BioRad) per BioRad protocol instructions, using 12.5 ng of cDNA for each well. Samples were run in duplicate and primers used in this study are shown in Table 3. Samples were run on the BioRad CFX thermocycler and data were quantified using the ΔΔCT method, as previously described[33]. All statistics were calculated using two-tailed $t$ tests.

**Immunofluorescence assays**. Cells were seeded in six-well plates on glass coverslips 24 h prior to each experiment and serum-starved for 4 h as described above. Cells were stimulated as specified in each figure legend and immediately washed with PBS and fixed in 4% paraformaldehyde for 10 min at room temperature. Cells were permeabilized for 10 min with 0.2% Triton-X 100 diluted in PBS and subsequently blocked with 2.5% BSA diluted in PBS (blocking solution) for 1 h. Cells were incubated with primary antibody diluted in blocking solution and incubated overnight at 4 °C (Table 1). Slides were washed three times for 5 min each with PBS and incubated with secondary antibodies diluted in blocking solution for 1 h at room temperature (Table 1). Cells were washed with PBS three times, 5 min per wash and incubated with DAPI (Table 1) diluted in PBS for 10 min at room temperature. Afterward, cells were washed with PBS and mounted onto glass slides using Fluoromount G. Slides were imaged by spinning disk confocal microscopy as specified in the figure legends. Different experimental samples were imaged using the same imaging settings each day. Colocalization was quantified using the ImageJ JACoP plugin. Specifics can be found in the figure legend.

**Statistics, reproducibility and data representation**. Throughout the manuscript all individual data points are plotted. Sample mean, median, and standard deviation are also shown on each bar graph. Unless noted in the figure legend, all statistical analyses were conducted using a two-tailed $t$ test and statistical significance denotes $p < 0.05$.

**Reporting summary**. Further information on research design is available in the Nature Research Reporting Summary linked to this article.

## Data availability
Supplementary Data 1 contains the Source data underlying Figs. 1b, c, f–h, 2d–g, 3c, d, f, 4c, e, h, 5a, b, and Supplementary Figs. 1b–d, 2b, e, 3b, c, 4b, c, and 5 a, b, d, f. The other datasets generated/analyzed during the current study are available from the corresponding author on reasonable request.

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

## Acknowledgements

M.S.D. thanks support from the National Science Foundation Graduate Research Fellowship and the University of Michigan Rackham Pre-doctoral Fellowship. The work is supported in part by Pardee Foundation, and a gift from Kendall and Susan Warren. M.S.D. would also like to thank Greg Thurber for allowing us to use his LiCor CLX imager for preliminary work in this study. M.S.D. would like to thank Manoj Puthenveedu for discussion and Wylie Stroberg for data interpretation feedback.

## Author contributions

M.S.D. conceived, designed, and performed all experiments, analyzed and interpreted the data, and wrote the manuscript. A.V.S. helped design and interpret experiments, and edited the manuscript. S.S. helped designed and interpret experiments and edited the manuscript. A.P.L. helped design and interpret/analyze experiments, and edited the manuscript.

## Competing interests

The authors declare no competing interests.
