## [Peer Review File · Communications Biology]

Reviewers' comments:

Reviewer #1 (Remarks to the Author):

This is a very interesting, original report on a third mechanism of intracellular GPCR signalling, using CXCR4 as a model. Evidence is provided that independent of membrane permeable ligands and endocytosis, upon stimulation, plasma membrane and internal pools of CXCR4 are post-translationally modified and collectively regulate EGR1 transcription. Together these data support a model where a small initial pool of plasma membrane-localised GPCRs are capable of activating internal receptor-dependent signalling events. β -arrestin-1 (arrestin 2) is necessary to mediate communication between plasma membrane and internal pools of CXCR4. These insights are of major importance as these may explain why CXCR4 over-expression is highly correlated with cancer metastasis and mortality, plasma membrane localisation is not, this will influence thinking in the field. The report can be of an additional value if the authors discuss the impact of their findings for CXCR4 drug development and or on the mechanisms of action on currently available CXCR4 inhibitors.

The data are presented clearly and methods are described in detail. Therefore, this paper adds valuable knowledge to the CXCL12/CXCR4 field.

Reviewer #2 (Remarks to the Author):

In the current manuscript, authors provide new evidence for GPCR signaling investigating the crosstalk between plasma membrane and internal pools of the chemokine receptor CXCR4. To this aim, they overexpressed CXCR4 (wt or the mutant K3R that showed a reduced (50%) plasma membrane targeting) in the RPE cell line. As marker of CXCR4 ligand-induced activation, authors used the commercially available antibody UMB2 that fails to detect CXCR4 upon CXCL12 stimulation due to receptor PTM.

They showed that independent of ligand cell entry, plasma membrane and intracellular CXCR4 pools are post-translationally modified upon CXCL12 stimulation. Authors identified β -arrestin 1 as the key molecular player that regulates the communication between plasma membrane and internal CXCR4 signaling finally controlling CXCL12-dependent EGR1 transcription.

The possibility that intracellular pools of CXCR4 can sustain signaling upon ligand triggering in cells that do express low level of plasma membrane receptor is partially new. Although several mechanisms remain to be addressed, this work provided preliminary but interesting findings on the topic. However, several points should be addressed before publication:

- At first, authors attempted to identify the specific PTM responsible for the loss of CXCR4 detection by the commercial UMB2 antibody. Nevertheless, they failed to restore UMB2 antibody level even after dephosphorylation of sample as indicated in the commercial data sheet. Please better comment on that. They concluded that the agonist dependent reduction in UMB2 detection was likely due to a combination of PTM. Did they test combined PTM inhibitors? Or other approach to identify PTM?
- Authors stated that "K3R mutant receptors partially colocalized with a Golgi compartment marker". How did they perform the colocalization analysis? Please provide fluorescence quantification and a colocalization index
- Authors suggested that CXCR4 is also present in other compartments than Golgi. Please, add additional markers for other intracellular compartments as ER, nucleus to further characterize CXCR4 intracellular localization. This will help to identify which intracellular CXCR4 pool is responsible for the intracellular signaling.
- Authors claimed that this study would help in resolving the paradox that while CXCR4 overexpression is associated with metastatic potential, plasma membrane localization is not. Did they test their hypothesis in vitro in tumor metastatic cell lines?

- Add statistic description to the Methods.

Importantly, authors need to revise the main text and figure legends providing additional experimental details to improve understanding.

Reviewer #3 (Remarks to the Author):

The manuscript entitled “ β -arrestin mediates communication between plasma membrane and intracellular GPCRs to regulate signaling” claims to identify a third, novel mechanism of GPCR signaling. The authors state that upon stimulation with receptor agonist not only plasma membrane-localized, but also intercellular CXCR4 molecules are post-translationally modified and regulate EGR1 transcription. The paper suggests that a small pool of plasma membrane-localized GPCRs can activate internal receptor-dependent signaling, and that β -arrestin-1 mediates this activation.

The findings appear original and significant. However, they appear to be of hypothetical nature as all the data is “consistent” with the hypothesis rather than proves it. Particular weakness is in the use of mutated receptor, in which three Lys residues in the C-terminal region have been replaced with Arg or two with Arg and one with Gln. Mutations lead to reduced surface expression of CXCR4 that authors heavily rely on for modulation of receptor localization. However, although Arg and Lys bare the same charge, these are not equivalent residues when it comes to protein-protein interactions and protein folding. Consequently, mutations most likely have affected many other signaling events and the data needs to be analyzed and discussed keeping this in mind.

Response to Reviewers

We thank the reviewers for their suggestions and constructive comments and have addressed them below, to the best of our knowledge, in a pointwise manner. The original comments of the reviewers are in black text, while our responses to those comments are in red font. The changes made in response to these comments are highlighted yellow in the revised manuscript.

Reviewer #1 (Remarks to the Author):

This is a very interesting, original report on a third mechanism of intracellular GPCR signalling, using CXCR4 as a model. Evidence is provided that independent of membrane permeable ligands and endocytosis, upon stimulation, plasma membrane and internal pools of CXCR4 are post-translationally modified and collectively regulate EGR1 transcription. Together these data support a model where a small initial pool of plasma membrane- localised GPCRs are capable of activating internal receptor-dependent signalling events. β -arrestin-1 (arrestin 2) is necessary to mediate communication between plasma membrane and internal pools of CXCR4. These insights are of major importance as these may explain why CXCR4 over-expression is highly correlated with cancer metastasis and mortality, plasma membrane localisation is not, this will influence thinking in the field. The report can be of an additional value if the authors discuss the impact of their findings for CXCR4 drug development and or on the mechanisms of action on currently available CXCR4 inhibitors. The data are presented clearly and methods are described in detail. Therefore, this paper adds valuable knowledge to the CXCL12/CXCR4 field.

Response: We would like to thank the reviewer for their thoughtful comments. We have edited the manuscript to include a greater discussion of the implications of intracellular CXCR4 activation and how this relates to drug development. We also briefly reviewed some of the therapeutic approaches to modulate CXCR4 function and comment on how localized GPCR signaling may alter/build upon these strategies.

Reviewer #2 (Remarks to the Author):

In the current manuscript, authors provide new evidence for GPCR signaling investigating the crosstalk between plasma membrane and internal pools of the chemokine receptor CXCR4. To this aim, they overexpressed CXCR4 (wt or the mutant K3R that showed a reduced (50%) plasma membrane targeting) in the RPE cell line. As marker of CXCR4 ligand-induced activation, authors used the commercially available antibody UMB2 that fails to detect CXCR4 upon CXCL12 stimulation due to receptor PTM. They showed that independent of ligand cell entry, plasma membrane and intracellular CXCR4 pools are post-translationally modified upon CXCL12 stimulation. Authors identified β -arrestin 1 as the key molecular player that regulates the communication between plasma membrane and internal CXCR4 signaling finally controlling CXCL12-dependent EGR1 transcription.

The possibility that intracellular pools of CXCR4 can sustain signaling upon ligand triggering in cells that do express low level of plasma membrane receptor is partially new. Although several mechanisms remain to be addressed, this work provided preliminary but interesting findings on the topic. However, several points should be addressed before publication:

At first, authors attempted to identify the specific PTM responsible for the loss of CXCR4 detection by the commercial UMB2 antibody. Nevertheless, they failed to restore UMB2 antibody level even after dephosphorylation of sample as indicated in the commercial data sheet. Please better comment on that. They concluded that the agonist dependent reduction in UMB2 detection was likely due to a combination of PTM. Did they test combined PTM inhibitors? Or other approach to identify PTM?

Response: We would like to thank the reviewer for their insightful comments regarding the identity of the detected CXCR4 PTMs. We completely agree with these points and wish that we were able to identify the specific PTM/combination of PTMs responsible for the agonist-induced decrease in UMB2 detection. As illustrated in Supplemental Fig. 1a-c, we attempted to rescue UMB2 detection using multiple inhibitors and receptor mutant combinations. Supplemental Fig 1a,b illustrates that phosphatase treatment alone is unable to rescue agonist-induced decrease in UMB2 detection. To rule out receptor ubiquitination and a combination of ubiquitination and phosphorylation, we conducted the same experiment with the ubiquitination-deficient mutant receptor K3R/Q. Again, irrespective of the receptor and phosphatase treatment, the UMB2 antibody failed to detect more CXCR4. AKT phosphorylation was used in both experiments as a phosphatase control and all lysates were collected without the addition of phosphatase inhibitors. These data suggest ubiquitination or phosphorylation alone does not lead to the agonist-induced decrease in UMB2 detection. It is important to note that this does not rule out CXCR4 ubiquitination at non-lysine residues or that CXCR4 phosphorylation is resistant to alkaline phosphatase and lambda phosphatase treatment (both on-blot and in-tube phosphatase treatment protocols were tested).

We are also aware of the datasheet statement regarding CXCR4 phosphorylation. However, as seen in Supplemental Fig. 1a,b, we were not able to replicate this finding. Therefore we decided to be transparent with reporting out findings. In addition to data shown in Supplemental Figure 1 (on-blot phosphatase treatment), we also tested the phosphatase treatment in-tube (proteins were lysed without phosphatase inhibitor), and adding NaOH (100 mM) pretreatment to attempt to remove non-lysine ubiquitination, and 100 mM beta-mercaptoethanol to attempt to disrupt any agonist-induced CXCR4 structural changes. None of these treatments restored UMB2 detection. Additionally, it has been previously reported that CXCR4 is potentially methylated in the binding region of the UMB2 antibody. To test this we used a methylation inhibitor (MTA – 5'-deoxy-5'-(methylthio)adenosine) and found no change in UMB2 detection upon agonist addition. We also attempted to use mass spectrometry to identify any unexpected PTMs located in the UMB2 antibody binding region. Unfortunately, we were only able to identify several already confirmed CXCR4 phosphorylation events. Due to limitations with protease cleavage sites in the C-terminus of CXCR4 and difficulties conducting mass spectrometry, our spectral counts for C-terminal residues had low peptide-spectrum matches (PSMs; <10) and contained multiple potential sites of PTM (ubiquitination, phosphorylation, methylation), which made PTM combinations difficult to decipher with our current approach. Ultimately, we decided that while identification of the specific PTMs would add to our conclusions, identification of the precise nature of PTM was not necessary to share our finding that intracellular CXCR4 are post-translationally modified. However, this is definitely an area that we believe warrants further investigation.

- Authors stated that “K3R mutant receptors partially colocalized with a Golgi compartment marker”. How did they perform the colocalization analysis? Please provide fluorescence quantification and a colocalization index

Response: We would like to thank the reviewer for this comment. In the updated manuscript we quantified WT and K3R receptor colocalization with GM130 (Golgi compartment marker) using Pearson’s correlation coefficient as well as line scan analysis. A description of the results is included in the updated manuscript and additional raw and quantified data are displayed in the updated Supplemental Fig 2.

- Authors suggested that CXCR4 is also present in other compartments than Golgi. Please, add additional markers for other intracellular compartments as ER, nucleus to further characterize CXCR4 intracellular localization. This will help to identify which intracellular CXCR4 pool is responsible for the intracellular signaling.

Response: We thank the reviewer for the recommendation. This was also something that we considered prior to our initial submission however; due to COVID-19 we were unable to conduct these experiments at the time. In the updated manuscript we have explored CXCR4 colocalization with early endosome marker EEA1, late endosome marker Rab7. We also conducted experiments with ER marker calreticulin. However, due to the diffuse labeling pattern of calreticulin, interpretation of calreticulin-CXCR4 colocalization was confounding. Interestingly we did observe partial colocalization between CXCR4 and EEA1 and Rab7 and these results are illustrated and quantified in Supplemental Fig. 2c-e. Results from these experiments have also been reported in the updated manuscript main text.

- Authors claimed that this study would help in resolving the paradox that while CXCR4 overexpression is associated with metastatic potential, plasma membrane localization is not. Did they test their hypothesis in vitro in tumor metastatic cell lines?

Response: We would like to thank the reviewer for the recommendation. We specifically chose to conduct the majority of this work in our CXCR4 overexpression model in order to be able to reliably monitor both post-translationally modified CXCR4 and total CXCR4 using an epitope tag (FLAG or MYC). To test our findings in a metastatic cancer cell line we conducted flow cytometry experiments in HeLa cells (Fig. 2e-g). We additionally attempted to look at agonist-induced CXCR4 PTMs in MDA231 cells however, were unable to detect CXCR4 using the UMB2 antibody by western blotting.

Our goal was to mechanistically investigate how CXCR4 localization impacts CXCR4 signaling and PTM. Given our resources, we believe that our current approach (overexpression model) provides the best controlled experiments that we could have used to explore these questions.

- Add statistical description to the Methods. Importantly, authors need to revise the main text and figure legends providing additional experimental details to improve understanding.

Response: We thank the reviewer for the comment. We have updated the manuscript, methods, and figure legends to improve experimental clarity.

Reviewer #3 (Remarks to the Author):

The manuscript entitled “ β -arrestin mediates communication between plasma membrane and intracellular GPCRs to regulate signaling” claims to identify a third, novel mechanism of GPCR signaling. The authors state that upon stimulation with receptor agonist not only plasma membrane-localized, but also intercellular CXCR4 molecules are post-translationally modified and regulate EGR1 transcription. The paper suggests that a small pool of plasma membrane-localized GPCRs can activate internal receptor-dependent signaling, and that β -arrestin-1 mediates this activation.

The findings appear original and significant. However, they appear to be of hypothetical nature as all the data is “consistent” with the hypothesis rather than proves it. Particular weakness is in the use of mutated receptor, in which three Lys residues in the C-terminal region have been replaced with Arg or two with Arg and one with Gln. Mutations lead to reduced surface expression of CXCR4 that authors heavily rely on for modulation of receptor localization. However, although Arg and Lys bare the same charge, these are not equivalent residues when it comes to protein-protein interactions and protein folding. Consequently, mutations most likely have affected many other signaling events and the data needs to be analyzed and discussed keeping this in mind.

Response: We would like to sincerely thank the reviewer for their insightful comments. We were also concerned by the possibility that Arg and Gln mutations are not identical and could impact CXCR4 signaling. To overcome this limitation, we repeated all experiments in Fig. 1 with a WT receptor that has a similar ~50% reduction in plasma membrane expression. To do so we removed a commonly used plasma membrane localization sequence on the N-terminus of WT CXCR4. This allowed us to reduce CXCR4 plasma membrane expression (~50%) while not mutating biologically relevant residues. Next we repeated the UMB2 detection and agonist-induced ERK1/2 and AKT signaling experiments comparing the high and low plasma membrane WT CXCR4. These findings are illustrated in Supplemental Fig. 5. Similarly to our findings with the K3R and K3R/Q mutants, WT CXCR4 plasma membrane expression did not change UMB2 detection (over the entire time course), and agonist-induced ERK1/2 and AKT phosphorylation (Supplemental Fig. 5). To avoid issues with the biologically relevant K3R and K3R/Q mutants, we used the WT receptors (with high and low plasma membrane expression confirmed in Supplemental Fig. 5) for the agonist-induced *EGR1* transcription experiments (Fig. 5). We have clarified this in the main text, as admittedly these important experimental distinctions were not clear. This has been updated in the manuscript and Fig. 5 figure legend. Additionally, we have added commentary about how Arg and Gln mutagenesis may impact CXCR4 signaling in an undetermined manner to better motivate the use of the high and low plasma membrane WT CXCR4 receptors for the agonist-induced *EGR1* transcription experiments.

We also agree with the reviewer’s comment that the presented data is preliminary and that the mechanism of intracellular CXCR4 PTM remains unclear. This is an active area of ongoing research in our laboratory. In particular we are interested in using localized $G_{\beta\gamma}$ inhibitors and

mini-G proteins to see whether we can detect CXCR4 activation at intracellular compartments.
This research was paused due to COVID-19.

REVIEWERS' COMMENTS:

Reviewer #2 (Remarks to the Author):

I appreciate the attention the authors have made to address all of the points I have brought up. Overall, authors have addressed my concerns sufficiently.

Reviewer #3 (Remarks to the Author):

The authors appeared to address most of the critique in the revised manuscript. Because major findings are original and significant, the manuscript can be published as is.